# Non-Toxigenic *Clostridioides difficile* Strain E4 (NTCD-E4) Prevents Establishment of Primary *C. difficile* Infection by Epidemic PCR Ribotype 027 in an In Vitro Human Gut Model

**DOI:** 10.3390/antibiotics12030435

**Published:** 2023-02-22

**Authors:** Perezimor Etifa, César Rodríguez, Céline Harmanus, Ingrid M. J. G. Sanders, Igor A. Sidorov, Olufunmilayo A. Mohammed, Emily Savage, Andrew R. Timms, Jane Freeman, Wiep Klaas Smits, Mark H. Wilcox, Simon D. Baines

**Affiliations:** 1Department of Food and Nutritional Sciences, School of Chemistry, Food and Pharmacy, Reading RG6 6DZ, UK; 2Facultad de Microbiología & CIET, Universidad de Costa Rica, San Pedro 11501-2060, Costa Rica; 3Leiden University Medical Center, Department of Medical Microbiology, Albinusdreef, P.O. Box 9600, 2300 RC Leiden, The Netherlands; 4Department of Clinical, Pharmaceutical and Biological Sciences, School of Life and Medical Sciences, University of Hertfordshire, Hatfield AL10 9AB, UK; 5Healthcare Associated Infections Research Group, Leeds Institute of Medical Research, University of Leeds, Leeds LS2 9JT, UK; 6Department of Microbiology, Leeds Teaching Hospitals NHS Trust, Leeds LS1 3EX, UK; 7Centre for Microbial Cell Biology, Einsteinweg 55, 2333 CC Leiden, The Netherlands

**Keywords:** *Clostridioides difficile*, RT027, non-toxigenic, antibiotics, resistance, gut model, colonisation, infection

## Abstract

*Clostridioides difficile* infection (CDI) remains a significant healthcare burden. Non-toxigenic *C. difficile* (NTCD) strains have shown a benefit in preventing porcine enteritis and in human recurrent CDI. In this study, we evaluated the efficacy of metronidazole-resistant NTCD-E4 in preventing CDI facilitated by a range of antimicrobials in an in vitro human gut model. NTCD-E4 spores (at a dose of 10^7^) were instilled 7 days before a clinical ribotype (RT) 027 (at the same dose) strain (210). In separate experiments, four different antimicrobials were used to perturb gut microbiotas; bacterial populations and cytotoxin production were determined using viable counting and Vero cell cytotoxicity, respectively. RT027 and NTCD-E4 proliferated in the in vitro model when inoculated singly, with RT027 demonstrating high-level cytotoxin (3-5-log_10_-relative units) production. In experiments where the gut model was pre-inoculated with NTCD-E4, RT027 was remained quiescent and failed to produce cytotoxins. NTCD-E4 showed mutations in *hsmA* and a gene homologous to CD196-1331, previously linked to medium-dependent metronidazole resistance, but lacked other metronidazole resistance determinants. This study showed that RT027 was unable to elicit simulated infection in the presence of NTCD-E4 following stimulation by four different antimicrobials. These data complement animal and clinical studies in suggesting NTCD offer prophylactic potential in the management of human CDI.

## 1. Introduction

*Clostridioides difficile* is an anaerobic, spore-forming, Gram-positive enteropathogen that can cause *C. difficile* infection (CDI) following the disruption of the indigenous gut microbiota by antimicrobial agents [1,2,3]. The perturbed microbiota allows *C. difficile* spore germination, cell outgrowth, and toxin production. CDI is the most serious cause of antibiotic-mediated diarrhea and leads to significant morbidity and mortality globally. According to a healthcare-associated infections survey undertaken by the European Centre for Disease Prevention and Control (ECDC) [4], the number of CDI cases in Europe from 2011–2012 was approximately 124,000. Furthermore, following a *C. difficile* surveillance study carried out by the ECDC in 2016, involving 24 million patient-days, 556 hospitals covering 20 countries, 7711 CDI cases were reported [5]. The financial burden of CDI on health institutions in Europe and the USA is estimated to be €3 billion and $4.8 billion, respectively [6,7]. 

CDI symptoms can vary from mild diarrhea, life-threatening pseudomembranous colitis, to death. Community infection is increasingly being reported, despite its history of recognition as a healthcare infection [8]. 

The current mainstay of treatment for CDI is antimicrobial therapy [9], with vancomycin and fidaxomicin the currently recommended first-line treatment options [10,11]; however, approximately 2 out of every 10 patients suffer recurrent CDI (rCDI) upon treatment of the primary episode and a further 40–65% upon treatment of a second episode [12]. This is particularly true with epidemic ribotypes such as 027 (NAP1/BI). Given the heightened failure rate, and the burden of CDI, new treatment options are being pursued. Fecal microbiota transplantation (FMT) of fecal material from a healthy donor to patients in order to replenish the perturbed gut microbiota which can resist *C. difficile* proliferation has been demonstrated to have a success rate of up to 95% in a number of well-designed studies [13,14]. However, FMT has been associated with serious adverse events such as aspiration pneumonia and the transfer of resistant organisms [15,16], and the long-term consequences of FMT are currently poorly understood. 

Three protein toxins namely toxin A (TcdA), toxin B (TcdB), and binary toxin (CDT) can be produced by virulent *C. difficile*, with TcdA/B being the chief virulence factors and primary drivers of symptoms, although CDT has been associated with increased severity of CDI [17,18]. The genes that encode TcdA (*tcdA*) and TcdB (*tcdB*) in toxigenic *C. difficile* strains are situated on a pathogenicity locus (PaLoc). In non-toxigenic *C. difficile* (NTCD) strains, the PaLoc is generally replaced by a non-coding 115/75-bp region [19]. 

NTCD strains have been shown to be efficacious and safe in preventing CDI in hamster models [20]. RT010 is a common *C. difficile* NTCD ribotype and lacks a functional PaLoc. Nagaro et al. [21] demonstrated in a hamster model that non-toxigenic strains prevented colonization by two toxigenic *C. difficile* (BI1 and BI6). A likely mechanism through which *C. difficile* strains compete for colonization in the gut is differences in their capacity to adhere to mucosal cells of the colon, as well as their different abilities to utilize limited essential nutrients [22]. NTCD strains have also been shown to be effective against rCDI in human trials [23,24]. These studies generally only investigated the effects of NTCD in relation to a single CDI-inciting antimicrobial. Few studies have evaluated the potency of NTCD in preventing primary CDI. 

In the present study, we assessed the efficacy of a NTCD isolate (NTCD-E4, ribotype 010) in preventing simulated CDI by the epidemic RT027 (strain 210, NAP1/BI) in a well-validated triple-stage in vitro human gut model in response to a wide range of CDI-inciting antimicrobials. The gut model mimics the microbiological and physicochemical conditions of the large intestine from proximal to distal and allows the study of the gut microbiota and pathogens and their response to antimicrobial agents. 

## 2. Materials and Methods

### 2.1. Clostridioides difficile Strains

Two *C. difficile* strains were used in this study. The first strain was a non-toxigenic *C. difficile* isolate, NTCD-E4, belonging to RT010. NTCD-E4 was originally isolated from the Leeds General Infirmary in 2001 from the environment of a care-for-the-elderly ward, as part of a longitudinal molecular epidemiology study of *C. difficile* infection [25] and has reduced susceptibility to metronidazole (MIC 8–32 mg/L). The strain was typed in detail as follows. In short, capillary PCR ribotyping [26] and a multiplex PCR targeting the 16S rRNA gene, *gluD*, and the genes encoding the large clostridial toxins and binary toxins according to recommendations by the European Center for Disease Control and Prevention [27] were performed at the Netherlands Reference Laboratory of *C. difficile*, hosted at the Leiden University Medical Center, The Netherlands. Metronidazole minimal inhibitory concentrations were determined using the agar dilution method according to CLSI guidelines on fresh Brucella Blood Agar (BBA) supplemented with 5 mg L^−1^ hemin and 1 mg L^−1^ vitamin K [28] with the EUCAST epidemiological cutoff of 2 mg L^−1^ to define metronidazole resistance in *C. difficile* [29]. PCR was used to establish the presence or absence of the metronidazole resistance-associated plasmid pCD-METRO as described previously [30]. The sequence of the *hsmA* gene, variants of which are associated with medium-dependent metronidazole resistance-associated in PCR ribotype 010 [9,31], and a putative pyridoxamine 5′-phosphate oxidase gene (homologous to CD196–1331) associated with metronidazole resistance in a genome-wide association study across multiple ribotypes [31] was analyzed as described before [9].

The second *C. difficile* strain studied was strain 210 (RT027, NAP1/BI, supplied courtesy of Dr. Rob Owens) which has been evaluated in multiple gut model experiments previously and was originally isolated from the Maine Medical Centre (Portland, OR, USA) and was one of the original NAP1/BI isolates that was implicated in severe outbreaks of CDI the USA in 2005 and subsequently worldwide and is therefore of significant clinical relevance. *C. difficile* strains were cultured on Brazier’s agar (Neogen, UK) from spore suspensions in order to produce viable cultures for MIC experiments and to inoculate agar plates for gut model spore suspensions. 

Antimicrobials were evaluated in separate mixed/competition *C. difficile* studies alongside controls in which NTCD-E4 or RT027 were singly inoculated, with dosing regimens following standard clinical dosing (frequency and duration) at gut-reflective antimicrobial concentrations. Concentrations of antimicrobials were DA, clindamycin 33.9 mg/L 6-hourly; CFX, cefotaxime 20 mg/L 12-hourly; CIP, ciprofloxacin 139 mg/L 12-hourly, AMP, ampicillin 8 mg/L 8-hourly [32,33,34,35]. Antimicrobial agents were evaluated over a 7-day instillation period (Figure 1). 

### 2.2. Triple Stage Chemostat Gut Model

We have previously described the use of a triple stage chemostat human gut model to study the interplay between *C. difficile,* antimicrobial agents, and the normal microbiota [36,37,38,39,40,41]. The gut model has proven to be a powerful and clinically reflective tool for the study of *C. difficile* infection and was validated against the physicochemical and microbiological measurements from the intestinal contents of sudden death victims [42]. Although gut-reflective in terms of these parameters, the system is limited by its inability to model immunological and secretory events within the colon. The gut model is comprised of three top-fed interconnected fermentation vessels and a dilution rate of 0.015 h^−1^ (retention time 66.7 h). The model is able to mimic the pH and nutritional availability from the proximal (Vessel 1, pH 5.5 ± 0.1), transverse (Vessel 2, pH 6.2 ± 0.1), and distal (Vessel 3, pH 6.8 ± 0.1) regions of the colon. The constituents and preparation of the growth medium are as previously reported [43].

### 2.3. Antimicrobial Susceptibility Testing

Antimicrobial susceptibility testing for the antimicrobials cefotaxime, ciprofloxacin, clindamycin, ampicillin, and tetracycline was performed using a modified Wilkins-Chalgren agar incorporation method as previously described [28], in order that Brazier’s breakpoint agars could be produced to differentiate between NTCD-E4 and RT027, total viable counts and spore counts in mixed culture. Antimicrobials were dissolved in deionized water, except for tetracycline (50% ethanol). *C. difficile* ATCC®700057 was used as an experimental control.

### 2.4. Enumeration of Gut Microbiota and C. difficile 

Total viable counts (TVC) of microbial populations enumerated included: total anaerobes (fastidious anaerobe agar), *Bacteroides fragilis* group (*Bacteroides* bile aesculin agar), *Bifidobacterium* spp. (Beeren’s agar), *Lactobacillus* spp. (LAMVAB agar), *Enterococcus* spp. (kanamycin aesculin azide agar), total facultative anaerobes (nutrient agar), and lactose-fermenting Enterobacteriaceae (MacConkey agar number 3) as previously described using a modified Miles and Misra viable counting technique [43]. *C. difficile* total viable counts were determined on Brazier’s agar CCEY incorporating 2% lysed horse blood and 0.5 mg/L tetracycline (RT027) or 10 mg/L clindamycin (NTCD-E4). During the course of gut model experiments, cultured *C. difficile* were subjected to cytotoxin testing (see below) to ensure that Brazier’s breakpoint agar isolates were of the correct phenotype. Agars were quality controlled after media preparation, following inoculation with cultures of the two *C. difficile* isolates. *C. difficile* spore counts were determined on the same Brazier’s breakpoint agars following alcohol-shock of samples at ambient temperature for 1 h. *C. difficile* TVCs that were enumerated without alcohol shock, comprise the cumulative vegetative and spore populations of *C. difficile* at a given time point. 

### 2.5. C. difficile Cytotoxin Assay

Cytotoxin production in gut model cultures was semi-quantified using a cell culture cytotoxicity assay with Vero cells (African Green Monkey kidney) as previously described [43]. In order to confirm the atoxigenic/toxigenic phenotype of NTCD/RT027, prior to the commencement of gut model experiments, NTCD-E4 and RT027 were anaerobically (Don Whitley Scientific, Bingley, UK) cultured for 72 h in brain heart infusion broth. Cultures were centrifuged at 16,000 × *g* for 5 min, and the culture supernatants were serially diluted to 10^−2^, with parallel neutralization with a *Clostridium sordellii* anti-toxin (Prolab, UK). Cytotoxin activity was evident as rounded Vero cells accompanied by neutralization of effect by the *C. sordellii* antitoxin. Cytotoxin titers (Relative Units, RU) were defined as the log_10_-reciprocal of the sample dilution where 80% cell rounding was observed.

### 2.6. Experimental Design

#### Whole Genome Sequencing and Analysis

Overnight brain heart infusion broth cultures of NTCD-E4 were centrifuged at 16,000 rpm for 2 min to pellet the cells. Cells were treated with lysozyme and proteinase K, and DNA was extracted using the Promega Wizard DNA extraction kit. The purity and quantity of DNA were detected using Nanodrop (Thermo Fisher Scientific, Loughborough, UK) and Qubit fluorimeter (Invitrogen Life Technologies, Paisley, UK). The DNA was then sent for sequencing, using a Pacific Biosciences RSII sequencer along with subsequent genome assembly and annotation at the Earlham Institute (Norwich Science Park, University of East Anglia, Norwich UK). PacBio reads were assembled into a single unit at the Earlham Institute. To rule out contamination, this assembly was analyzed using Kraken2 [44] and the MiniKraken DB 4 Gb, which contains complete bacterial genomes deposited in RefSeq. In addition, genome completeness and heterogeneity were checked using CheckM [45]. The phenotypic identification of strain NTCD-E4 as *C. difficile* was confirmed using rMLST [46], TYGS [47], and MiGA [48]. Furthermore, its sequence type was determined with FastMLST [49]. An abricate-based search against the CARD database [50] was done to detect antibiotic-resistant genes. Upon annotation with Bakta [51], the resulting gff3 file was examined visually to confirm the lack of PaLoc genes. 

## 3. Results

### 3.1. Antimicrobial Susceptibility Testing

MICs were screened for ciprofloxacin, cefotaxime, ampicillin, clindamycin, and tetracycline using a Wilkins-Chalgren agar incorporation method [28,52], in order to inform on the most appropriate antimicrobials to incorporate into Brazier’s breakpoint agars and also the antimicrobials to study in gut model experiments. MICs for RT027 and NTCD are displayed in Table 1.

The MICs allowed for the preparation of breakpoint agars that distinguished between NTCD-E4 (Brazier’s agar supplemented with 10 mg/L of clindamycin) and RT027 (Brazier’s agar supplemented with 0.5 mg/L of tetracycline). *C. difficile* were routinely cultured on new batches of breakpoint agars to ensure the agars only grew the desired *C. difficile* strain. NTCD-E4 is Erm(B) positive as demonstrated by our WGS analysis, but the tetracycline resistance determinant(s) in RT027 were not determined in this study. 

### 3.2. Genome Analysis of NTCD-E4

De novo assembly of the total genomic DNA of strain NTCD-E4 returned a single contiguous chromosome and did not reveal evidence for extrachromosomal elements. Raw sequence data, as well as the assembled *C. difficile* NTCD-E4 genome sequence, is available under BioProject number PRJNA917292.

NTCD-E4 was unequivocally identified as *Clostridioides difficile* on account of its genome length (4.18 Mb), %GC content (28.6%), hits to *Clostridioides* 16S rRNA genes deposited in the RDP database (12 hits, 100% confidence), and digital DNA-DNA hybridization (93.8%), ANI (99.2%), and AAI (95%) to *C. difficile* ATCC 9689 (93.8%). Our in silico analyses assigned this strain to *C. difficile* ST15 (similar to other RT010 strains [9] and rMLST 24726, and we did not find evidence of carriage of PaLoc-associated genes or of bacteriocin biosynthetic pathways. 

A MiGA-based MyTaxa scan distinguished 13 regions with unusual taxonomic distribution in the genome of NTCD-E4. Nonetheless, Mobile Element Finder (db 1.0.2) did not detect mobile genetic elements, ConjScan MacSyFinder did not find Firmicutes/Actinobacteria- or Firmicutes/Actinobacteria/Tenericutes/Archaea conjugation systems, and ICEfinder only recognized one putative integrative and mobilizable element.

We used ABRicate to assess antimicrobial resistance determinants. This revealed, with 100% identity and coverage at the nucleotide level, two genes anticipated to play a role in resistance to streptomycin (ant(6)-Ia) and MLS_B_ antibiotics (*erm*(B)). Of note, ABRicate does not identify metronidazole resistance mechanisms of *C. difficile*, and considering the AST results we manually evaluated these. Metronidazole MIC > 8 mg/L is generally associated with the 7-kb plasmid pCD-METRO [30]. Though the de novo assembly of the long read sequences did not recover this plasmid, preparation of PacBio sequencing libraries generally includes a sizing step that can reduce the abundance of small plasmids. We, therefore, determined plasmid carriage using a PCR-based approach as described previously [30], but we found no evidence for the presence of this plasmid. Reflex AST testing of the isolate did confirm high-level resistance, which is uncommon for plasmid-negative isolates. Metronidazole resistance can also be related to specific chromosomal loci; most notably, a 1-bp deletion of the haem-inducible *hsmA* gene contains RT010 isolates with medium-dependent metronidazole resistance [9], and SNPs in the chromosomal region containing a pyridoxamine 5′-phosphate oxidase (CD196–1331 in strain CD196), as identified in a large genome-wide association study [53]. We determined the SNP signatures of these genes compared to the reference sequences from strains 630 and R20291. NTCD-E4 was found to contain the previously described 1-bp deletion in *hsmA*. Moreover, NTCD-E4 also contains two SNPs in the second chromosomal region. The first is an A > C mutation that results in a Tyr130Ser amino acid substitution, identical to what has been reported [54]. The second is an A > T mutation 30 bp upstream of the ATG start codon of the gene, which leads to the generation of a perfect TATAAT sequence and likely increases the expression of the putative resistance gene. We cannot confirm whether this is the same promoter SNP identified in the GWAS study, as this is not further specified in that manuscript. One or more of these SNPs likely explain the metronidazole resistance phenotype of NTCD-E4. Previous studies have also associated SNPs in genes corresponding to *feoB1*, *nifJ*, *xdh*, *iscR*, *fur*, CDR20291_0749, and CDR20291_2649 with metronidazole resistance [54,55,56], but no frameshift, synonymous, or non-synonymous mutations were identified in NTCD-E4 in these genes.

### 3.3. Gut Model Experiments

In order to assess the propensity of NTCD-E4 in preventing RT027 CDI in the gut model, we evaluated a range of antimicrobial agents as inducers of simulated CDI in separate gut model experiments. Data from all gut model vessels were determined; however, only data from V3 of the gut model will be presented in this manuscript given that severity of CDI pathophysiology is generally greater in the distal colon represented by vessel 3 (based on pH and nutrient availability).

Data generated from the competition experiments are displayed in the main body of this manuscript and data from singly inoculated gut model experiments are provided as Appendix A. We noted that in the absence of antimicrobial perturbation of the gut microbiota, both *C. difficile* strains remained quiescent as spores and no cytotoxin was detected in the competition gut model. 

The NTCD-E4 and RT027 strains had the same ampicillin MICs (Table 1). Ampicillin was evaluated in a competition gut model of CDI with both NTCD-E4 and RT027 (Figure 2A), and in separate gut model experiments as an inducer for simulated CDI with inoculation of NTCD-E4 (Appendix A) or RT027 singly inoculated (Appendix A). Germination and outgrowth of *C. difficile* spores were observed in both gut models when the strains were tested individually (Appendix A). RT027 spores germinated, and vegetative *C. difficile* populations were observed 4 days after ampicillin instillation, with cytotoxin production first detected 24 h later. RT027 appeared to undergo a second cycle of growth and further elevation of cytotoxin production 11 days after cessation of ampicillin instillation. Cytotoxin titers reached 3RU in the RT027 singly inoculated gut model (Appendix A). NTCD-E4 spores germinated after 7 days of ampicillin instillation, and marked vegetative populations remained until the end of the experiment with no cytotoxin production detected throughout. In the gut model, an experimental treatment where NTCD-E4 and RT027 co-competed, NTCD-E4 spores germinated and marked vegetative growth was observed 5 days following commencement of ampicillin instillation, whereas RT027 remained quiescent with no signs of spore germination, vegetative growth, or cytotoxin production. Ampicillin instillation adversely affected viable counts of obligate anaerobic gut microbiota; however, facultative anaerobe viable counts were unaffected or increased (Table 2).

### 3.4. Clindamycin as an Inducer of CDI

MICs of clindamycin against NTCD-E4 and RT027 differed substantially. NTCD-E4 demonstrated high-level resistance (MIC > 128 mg/L), significantly above the breakpoint of 8 mg/L [29], whereas RT027 was susceptible (MIC 2 mg/L) (Table 1). Genome analysis suggested that the resistance phenotype of NTCD-E4 is likely explained by the carriage of *erm(B).*


Clindamycin has previously been shown to induce simulated CDI by this RT027 strain in the gut model, and it is the antimicrobial of choice for inducing CDI in animal models [36,41]. Clindamycin was evaluated as an inducer for simulated CDI in a competition model of CDI with both NTCD-E4 and RT027 (Figure 2B), and in singly inoculated gut model experiments with NTCD-E4 (Appendix A) and RT027 (Appendix A). In the singly inoculated RT027 gut model (Appendix A), RT027 spores were activated 10 days after cessation of clindamycin instillation and the cytotoxin titers reached 4 RU. In the singly inoculated NTCD-E4 gut model, NTCD-E4 spores germinated on day 4 of clindamycin instillation, and high-level growth above spore counts persisted for the remainder of the experiment (Appendix A). In the competition gut model (Figure 2B) NTCD-E4 spores germinated, and high-level growth was observed in a similar fashion to that observed in the singly inoculated NTCD-E4 gut model, but RT027 spores were not visibly activated following clindamycin instillation, and cytotoxin was not detected throughout the experiment. Indeed, RT027 spores were infrequently detected in the gut model after day 38, indicating that the organism had been washed out of the gut model. The gut microbiota was markedly perturbed by clindamycin instillation, with declines in viable counts of members of obligate anaerobes and lactobacilli, with increased viable counts of enterococci and lactose-fermenting Enterobacteriaceae (Table 2).

### 3.5. Ciprofloxacin as an Inducer of CDI

MICs of ciprofloxacin against NTCD-E4 and RT027 differed 4-fold, with NTCD-E4 demonstrating susceptibility against a breakpoint of ≥8 mg/L [29], whereas RT027 was resistant (Table 1). 

Ciprofloxacin has been studied previously with RT027 as an inducer for simulated CDI in the gut model [39]. Ciprofloxacin was evaluated in a competition gut model of CDI with both NTCD-E4 and RT027 (Figure 2C), and in separate gut model experiments as an inducer for simulated CDI with NTCD-E4 alone (Appendix A), and RT027 alone (Appendix A), and RT027 spores were activated 7 days after the cessation of ciprofloxacin instillation and high-level cytotoxin production was observed (Appendix A, 5RU), along with substantial vegetative populations until the end of the experiment. NTCD-E4 spores germinated 4 days after cessation of ciprofloxacin instillation in both gut models containing this *C. difficile* strain, and vegetative cell proliferation was observed until the end of the experiment (Figure 2C and Appendix A). In the competition model, RT027 remained quiescent following ciprofloxacin-induced perturbation of the gut microbiota, and cytotoxin remained undetectable for the duration of the experiments. Ciprofloxacin instillation reduced the viable counts of all gut microbiota groups enumerated (Table 2). 

### 3.6. Cefotaxime as an Inducer of CDI

MICs of cefotaxime against NTCD-E4 and RT027 differed by only 2-fold, with both strains demonstrating high MICs (Table 1). The ability of cefotaxime to induce CDI has been previously shown for non-RT027 strains [1,50], but cefotaxime MICs are similar between RT027 and RT001. Cefotaxime was evaluated in separate gut model experiments as an inducer for simulated CDI with NTCD-E4 alone (Appendix A), RT027 alone (Appendix A), and in a competition experiment with both *C. difficile* strains (Figure 2D). When tested singly, RT027 spores were activated 6 days after the commencement of cefotaxime instillation (Appendix A), and a substantial period of vegetative growth was observed until the end of the experiment, along with peak cytotoxin titers of 3 RU. In the singly inoculated NTCD-E4 gut model experiment, spores remained quiescent prior to cefotaxime instillation, but NTCD-E4 spores germinated after 5 days of antimicrobial instillation, and substantial vegetative populations remained until the end of the experiment (Appendix A). In the competition gut model experiment where NTCD-E4 and RT027 competed following cefotaxime installation, NTCD-E4 demonstrated a similar spore germination and growth profile to that seen in the singly inoculated gut model (Appendix A), whereas RT027 failed to outgrow from spores, and cytotoxin was absent for the duration of the experiment (Figure 2D). Indeed, RT027 spores were rapidly washed out of the gut model during the period when NTCD-E4 was actively growing. Cefotaxime instillation deleteriously affected obligately anaerobic gut microbiota groups and lactose-fermenting Enterobacteriaceae, whereas enterococci and lactobacilli were unaffected or increased (Table 2). 

## 4. Discussion

*C. difficile* infection remains a healthcare burden worldwide and is generally considered to be a consequence of antimicrobial perturbation of the gut microbiota, which allows virulent *C. difficile* to opportunistically colonize the large intestine and cause CDI [3]. The recommended treatment options for CDI have narrowed in recent years [11,57] due to a lack of efficacy of metronidazole as a consequence of a poor pharmacokinetic profile and the emergence of reduced susceptibility/resistance [58,59,60]. Furthermore, currently effective antimicrobial treatments (vancomycin and fidaxomicin) can disrupt the gut microbiota, leading to the potential for recurrent CDI that requires further antimicrobial intervention or FMT to resolve. Thus, strategies to prevent CDI are still required. 

The present study evaluated the effectiveness of a NTCD strain, NTCD-E4, in preventing primary CDI by a clinical RT027 strain, using a complex in vitro human gut model. The RT027 isolate selected in this study was one of the original NAP1/BI isolates that were implicated in severe outbreaks of CDI in the USA in 2005 and subsequently worldwide, and they are, therefore, of significant clinical relevance. This isolate has been studied in multiple clinical and scientific CDI studies previously and produces high-level cytotoxins, bimodal growth, and high sporulation. Therefore, this isolate poses a significant challenge for the NTCD. This RT027 isolate has been studied extensively in the gut model previously [39,61] and, therefore, substantial data exists regarding its behavior in this clinically reflective in vitro environment. NTCD-E4 was selected due to its antimicrobial susceptibility profile allowing for it to be easily viably counted from a mixed culture of *C. difficile*, that is, RT027 mixed with NTCD-E4. It is the same ribotype (PCR ribotype 010) as NTCD-M3, investigated in hamster and human studies for the prevention of CDI [20,24]. NTCD-E4 is a well-studied isolate in our laboratories in terms of its sporulation, antimicrobial resistance, and genotypic profile. It was, therefore, judged a suitable strain to compete against RT027, with relevance to preceding clinical and animal studies of this PCR ribotype.

The genomic analysis of NTCD-E4 in this study may aid the assessment of genetic factors that explain why this NTCD strain was able to prevent CDI and also whether this strain might be suitable for wider in vivo studies in the future. The human gut model used in these studies is a complex continuous culture system designed to reflect the conditions within the large intestine and comprises three interconnected fermentation vessels at differing pH and nutrient availability. The epidemic RT027 strain selected for these competition studies has been studied extensively in the gut model previously, including work with a range of CDI-inducing antimicrobials [36,38,61,62], but not in the context of an NTCD intervention to date. 

NTCD strains offer significant potential as therapeutic interventions for CDI [20,21,24]. The success of NTCD in preventing CDI has been demonstrated previously in the hamster model [20,21], in neonatal piglets [63], and also in humans in the treatment of recurrent CDI [24]. Furthermore, NTCD has been demonstrated in Phase I clinical trials to be well tolerated and safe [64], and it is also able to significantly reduce recurrent CDI from 30% in placebo group versus 11% (10^4^ spore dose) and 5% (10^7^ spore dose) alongside concurrent antimicrobial therapy in a Phase II randomized clinical trial [24]. Indeed, significant colonization of humans with NTCD was also demonstrated, therefore a prophylactic biotherapeutic approach for prevention of CDI with NTCD is an attractive proposition. Moreover, NTCD strains can offer a non-toxigenic platform for the expression of immunogenic proteins that confer protection against *C. difficile* infection [65,66]. These studies are intriguing and complement the present study which employed a wild-type NTCD strain and multiple inducing antimicrobial agents in a well-validated human gut model. 

### 4.1. Antimicrobial Inducers of Simulated CDI 

Where previous studies in animals and humans focused on evaluating the potential of NTCD interventions relating to CDI resulting from perturbation by a single antimicrobial agent, the present study evaluated a broad range of antimicrobial agents to perturb the gut microbiota and to gain insight into the potential for NTCD-E4 to prevent simulated CDI in vitro. Supra-MIC of antimicrobials is expected to prevent *C. difficile* spore germination, proliferation, and cytotoxin production, and it is not until bioactive antimicrobial concentrations decline below the MIC against *C. difficile* that spores become activated [37,39,67,68]. 

The findings of this study were stark; NTCD-E4 prevented the RT027 strain tested from establishing CDI within the gut model following perturbation by all antimicrobial agents studied. Reduced microbiota populations of members of the *Bacteroides fragilis* group and *Bifidobacterium* spp. correlated with *C. difficile* spore germination, in line with prior gut model studies. When inoculated singly, NTCD-E4 behaved in a similar manner to other *C. difficile* ribotypes that were evaluated in the gut model; see [1,36,38,43,62,68] and our unpublished observations. Spores remained quiescent in the absence of antimicrobial perturbation and were washed out of the gut model, then spores developed into vegetative forms and proliferated at a higher specific growth rate than the dilution rate of the chemostat system, hence increasing the viable counts. In the competition gut models, NTCD-E4 spore germination and vegetative cell outgrowth and proliferation occurred sooner than RT027 would have been expected to elicit growth and cytotoxin production, based on the results from the singly inoculated RT027 gut model experiments, including with potent antimicrobial inducers such as fluoroquinolones that were a major driver for the global spread of this epidemic strain [69]. NTCD-E4 spore germination/outgrowth was observed shortly after the commencement of clindamycin instillation, likely due to its resistance to the antimicrobial (Table 1) as a consequence of possession of the *erm(B)* gene. 

### 4.2. Potential Mechanisms of Antagonism of Virulent C. difficile

The apparent broadly protective effect of NTCD-E4 against primary CDI in the gut model could result from multiple, not mutually exclusive, mechanisms such as antimicrobial resistance, nutrient competition and/or cross-feeding, and bacteriocin production. 

Based on the fact that protection is observed with all CDI-inciting antimicrobials and appears independent of the antimicrobial susceptibility of the isolates (see above), we consider this to be an unlikely mechanism. Nevertheless, NTCD-E4 has an antimicrobial susceptibility profile that may give it a competitive advantage over virulent *C. difficile* ribotypes under certain circumstances, for example, metronidazole reduced susceptibility, MLS_B_ resistance, and high-level cephalosporin resistance. Metronidazole resistance of NTCD-E4 is likely the result of SNPs in *hsmA* and/or a gene homologous to CD196_1331 (see above). Of note, a recent preprint provides further support for the causative involvement of, in particular, the latter mutation [70]. NTCD-E4 has other features that make it of interest as a strain for potential interventions. It does not contain plasmids, the components linked to metronidazole that reduced susceptibility/resistance are chromosomal, and this isolate is susceptible to both primary treatment antimicrobials, vancomycin, and fidaxomicin (unpublished data).

*C. difficile* utilizes an incredible array of nutrients and is uniquely adapted to colonization of the large intestine [71]. In alternative growth media, nutrients such as biotin, amino acids, and glucose have been demonstrated to affect *C. difficile* growth and toxin expression [72,73,74]. Additionally, the soluble CspC pseudoprotease has been suggested to regulate *C. difficile* spore germination in response to bile acids, amino acids, and calcium [75,76]. It remains to be established which, if any, nutrients might be limiting in the three-stage gut model employed here. It is likely that nutrient utilization profiles differ between *C. difficile* types, and this deserves to be investigated with larger numbers of strains. Nevertheless, RT027 was previously shown to possess an expanded nutritional utilization profile compared to RT013 and RT078; in particular, in its ability to utilize nitrogen sources [77]; for NTCD-E4 (or RT010 strains in general) this has not been assessed. Despite the broad nutrient utilization capability of RT027, we found that this RT could still not establish itself in the gut model when competing with NTCD-E4. This may indicate that NTCD-E4 has a greater affinity for key nutrient(s) within the gut environment, or alternatively, that NTCD-E4, once activated, consumed key nutrient(s) such that the bioavailable concentrations were below a threshold needed to support RT027 spore germination, outgrowth, and/or vegetative cell proliferation. Nutritional profiling in future gut model experiments may elucidate any changes in concentrations of bile acids, amino acids, or other germinant/co-germinants that may stimulate *C. difficile* spore germination once bioactive antimicrobial concentrations are no longer inhibitory. 

Bacteriocins can contribute to shaping a gut ecosystem, and differential susceptibility of the *C. difficile* strains to gut-microbiota-produced compounds might account for the differences in the growth kinetics observed here. For *C. difficile* specifically, R-type diffocins (bactericidal high-molecular-weight phage tail-like bacteriocins) have been demonstrated to be produced by some *C. difficile* isolates following the SOS response [78], but their role in the ecology of CDI remains to be determined in more complex models. Our bioinformatics analysis of NTCD-E4 in the present study did not highlight any putative bacteriocins that may specifically be responsible for the inhibition of RT027, but this should be confirmed by in vitro experimentation. 

As phylogenetically divergent NTCD strains have demonstrated in vitro and in vivo efficacy in CDI prevention [21,24,63,79,80], it remains a possibility that as of yet uncharacterized factors determine the efficacy of NTCD for the prevention of CDI by virulent strains. 

### 4.3. Risk of PaLoc Transfer to NTCD

One cause for caution in the use of NTCD as treatments for CDI as either prophylactic agents, or in the treatment of symptomatic CDI, is the potential for NTCD to be converted into toxigenic *C. difficile* following the acquisition of an intact pathogenicity locus [81]. PaLoc horizontal gene transfer, albeit at low frequency, has been observed in optimized laboratory conditions using *C. difficile* 630Δerm as a donor in mating with CD37 (RT009), RT138, and RT140 strains. Furthermore, PaLoc transfer has been suggested to occur in vivo in wild populations [82]. Phenotypic analysis of *C. difficile* isolated on selective media in the present study did not demonstrate any *C. difficile* isolates with an antibiogram reflective of NTCD-E4 that gained the ability to produce *C. difficile* cytotoxins. Though further work is needed and underway under optimized laboratory conditions to evaluate this possibility, our preliminary data are in line with a recent study that failed to identify PaLoc transfer to NTCD-M3 (a REA-type M strain, that includes RT010) under in vitro conditions [83].

## 5. Conclusions

The results of this study highlight the potential for NTCD as an intervention, potentially as a prophylactic oral agent (spores) in patients and/or animals at risk for CDI. This study uniquely demonstrated that NTCD-E4 was able to prevent RT027 CDI in response to four distinct CDI-inducing antimicrobial agents in a human gut model system, whereas most other studies use only a single inciting antimicrobial. If effective in the clinical setting, a NTCD intervention would significantly reduce antimicrobial consumption for CDI, reduce ongoing perturbation of the gut microbiota, and likely reduce the selection pressure for resistance development in CDI patients due to antimicrobial therapy.

Future work, using the gut model and other suitable experimental systems, is needed to understand strain-specific beneficial characteristics (fitness indices, specific growth rates, nutrient competition, spore germination kinetics), barriers to implementation of this approach, and mechanisms of preventing CDI by virulent strains and assess the risks associated with the potential transfer of antimicrobial resistance determinants as well as the PaLoc transfer between NTCD and virulent *C. difficile* strains.

## Figures and Tables

**Figure 1 antibiotics-12-00435-f001:**
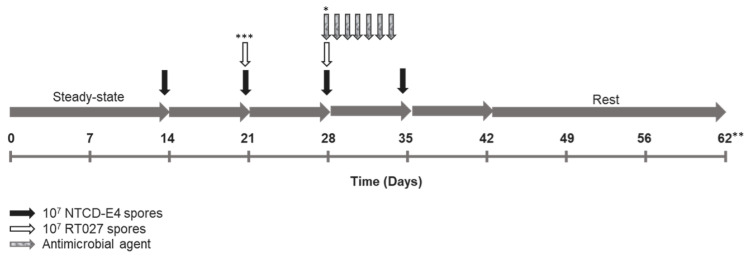
Experimental design for gut model studies evaluating non-toxigenic *Clostridioides difficile* (NTCD-E4) and *C. difficile* ribotype 027 (RT027). * Antimicrobial agents were evaluated individually over 7 days in competition *C. difficile* studies (gut models pre-inoculated with NTCD-E4) alongside controls of singly inoculated NTCD-E4 and RT027, with dosing regimens following standard clinical dosing (frequency and duration) at gut-reflective antimicrobial concentrations. DA, clindamycin 33.9 mg/L 6-hourly; CFX, cefotaxime 20 mg/L 12-hourly; CIP, ciprofloxacin 139 mg/L 12-hourly, AMP, ampicillin 8 mg/L 8-hourly [32,33,34,35]. ** CIP experiment was terminated after 56 days. *** In experiments with NTCD alone, an additional dose of NTCD-E4 spores was instilled on day 21 rather than RT027 spores.

**Figure 2 antibiotics-12-00435-f002:**
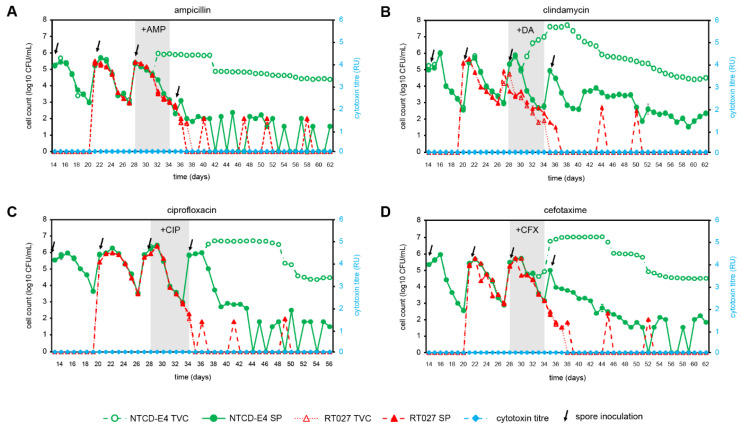
Effect of antimicrobial agents (ampicillin, AMP; clindamycin, DA; ciprofloxacin, CIP; cefotaxime, CFX) on *C. difficile* strains, NTCD-E4 and RT027 in a competition gut model of NTCD-E4 and RT027, total viable counts (TVC), spore counts (SP), and cytotoxin titres (RU) in a human gut model (Vessel 3, pH 6.8). Brazier’s agar was used to culture all *C. difficile* strains, incorporating tetracycline (TET, 0.5 mg/L) for RT027 and clindamycin (DA, 10 mg/L) for NTCD-E4. Spore counts were determined following an alcohol shock, and cytotoxin titers (relative units, RU) were determined using a Vero cell cytotoxicity assay. (**A**), AMP, 8 mg/L, 8-hourly; (**B**), DA, 33.9 mg/L, 6-hourly; (**C**). CIP, 139 mg/L, 12-hourly; (**D**), CFX, 20 mg/L, 12-hourly.

**Table 1 antibiotics-12-00435-t001:** Minimum inhibitory concentrations (MIC, mg/L) of antimicrobial agents using a Wilkins–Chalgren agar incorporation method. Bacterial inocula were cultured in Schaedler’s anaerobe broth and inoculated onto agar using a multi-point inoculator (10^4^ cfu/spot). CIP, ciprofloxacin; CFX, cefotaxime; AMP, ampicillin; DA, clindamycin; TET, tetracycline; MTZ, metronidazole.

	CIP	CFX	AMP	DA	TET	MTZ
NTCD-E4	2	128	4	>128	0.06	8
RT027	8	64	4	2	16	1

**Table 2 antibiotics-12-00435-t002:** Changes (log_10_cfu/mL) in viable counts of gut microbiota groups following exposure to antimicrobial agents in a human gut model.

Antimicrobial	*Bacteroides fragilis* Group	*Bifidobacterium* spp.	*Enterococcus* spp.	Facultative Anaerobes *Lactobacillus* spp.	Lactose Fermenting Enterobacteriaceae
Ampicillin	−3	−3	2	2	2
Clindamycin	−3	−4	2	2	3
Ciprofloxacin	−2	−1	−1	−1	−4
Cefotaxime	−2	−3	0	2	−1

## Data Availability

The ethical approval for this study does not permit the release of the supporting data in a public repository.

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
