# Peer review of "Non-Toxigenic Clostridioides difficile Strain E4 (NTCD-E4) Prevents Establishment of Primary C. difficile Infection by Epidemic PCR Ribotype 027 in an In Vitro Human Gut Model"

_antibiotics, 2023, doi:10.3390/antibiotics12030435_

Round 1

Reviewer 1 Report

Perezimor et al present data for use of an NTCD strain in the well-established gut model of CDI. I am a bit surprised at the study design which places both the NTCD strain and the toxigenic strain in the model at the same time in competition. Clinically it is hard to find examples that would replicate such a situation, and the results of these combined experiments document that one strain quickly dominates and forces out the second strain, in this case NTCD dominates following all 4 antibiotics tested despite differences in MIC that favor RT027 in some instances. Also of concern is the choice of a NTCD strain exhibiting unusual antibiotic resistance to metronidazole and clindamycin. Clinically use of strains bearing unusual resistance will raise concern about resistance transfer in the gut. Transferable ermB presence accounts for clindamycin resistance and the authors go to great lengths to determine the metronidazole resistance mechanism for which multiple known mechanisms are ruled out. What is most impressive is the manner in which NTCD dominates RT027 under all antibiotic challenges. It appears that a key finding is that spore germination of NTCD is faster than RT027 with all antibiotics (despite MIC differences) and this may be the factor that accounts for domination. How does NTCD do this and will it also do so with other toxigenic strains? Is NTCD-E4 unique in this dominance? Key questions requiring further study. NTCD has been considered a CDI preventive since the early 1980s work of Peter Borriello and is still not approved for use in humans by regulatory agencies. The observations of this study support its continued development.  My specific comments are below:

Line Number

48 While CDI is an important cause of AAD or AMD, it is not the main cause as it likely accounts for less than 25% of AAD. Suggest you change language to "CDI is the most serious cause".....

144 Was taurocholate or another bile acid added to the media to enhance spore germination?

356 For how many days were ampicillin and the other antibiotics given or was it a single day dosing?

579 This PaLoc transfer could not be replicated with NTCD-M3 as the recipient under the same in vitro conditions.

Sambol SP, Johnson S, Cheknis A, Gerding DN. Absence of toxin gene transfer from Clostridioides difficile strain 630deltaerm to nontoxigenic C. difficile strain NTCD-M3r in filter mating experiments. Plos One 2022 Jun 29;17(6):e0270119.

Discussion General: The Discussion is too long. There is no need to reiterate every prior Gut Model paper using prior antibiotics and strains in this model. I believe it could easily be reduced to half its current length by simply focusing on the key points of the paper. I would also like to see more speculation on the mechanism by which NTCD dominates in this model.

Author Response

General comments

Thank you for your in-depth review. This was helpful in refining this submission. The experimental design used was in order to assess the prophylactic potential of NTCD in preventing primary CDI, the idea being that prior dosing a patient/animal with NTCD may allow for antagonism of virulent strains prior to establishment of an infection, since the point at which acquisition of a virulent strain occurs in a patient/animal cannot be predicted. We discuss the antimicrobial resistance profile of NTCD-E4 in the manuscript and highlight that metronidazole reduced susceptibility was not transferred to RT027 in these studies and we are assessing the mechanism of reduced susceptibility further. ErmB is widespread in gut microbiota populations and is discussed in the manuscript but this highlights an interesting proposition that the presence of ErmB may provide a selective advantage to NTCD-E4.

Line Number

48 While CDI is an important cause of AAD or AMD, it is not the main cause as it likely accounts for less than 25% of AAD. Suggest you change language to "CDI is the most serious cause".....

Response – we accept this revision and have inserted this.

144 Was taurocholate or another bile acid added to the media to enhance spore germination?

Response – Brazier’s agar contains cholic acid to facilitate spore germination. We have tried additional supplementation with lysozyme and other bile acids but have not found consistent elevation in spore recovery, hence use the standard formula

356 For how many days were ampicillin and the other antibiotics given or was it a single day dosing?

Response – 7 days antimicrobial dosing was completed as per Figure 1 (details in the title) and Line 115. I have amended the wording to clarify this to:

Antimicrobial agents were evaluated over a 7 day instillation period, in separate…..

579 This PaLoc transfer could not be replicated with NTCD-M3 as the recipient under the same in vitro conditions.

Sambol SP, Johnson S, Cheknis A, Gerding DN. Absence of toxin gene transfer from Clostridioides difficile strain 630deltaerm to nontoxigenic C. difficile strain NTCD-M3r in filter mating experiments. Plos One 2022 Jun 29;17(6):e0270119.

Response – thank you for suggesting this inclusion. This has been incorporated into the revised manuscript.

Discussion General: The Discussion is too long. There is no need to reiterate every prior Gut Model paper using prior antibiotics and strains in this model. I believe it could easily be reduced to half its current length by simply focusing on the key points of the paper. I would also like to see more speculation on the mechanism by which NTCD dominates in this model.

Response – Thanks for these comments. The initial discussion was prepared to analyse the fact that NTCD have not been assessed in these experimental conditions previously, with this array of antimicrobial inducers for CDI, hence this involved a comparison to prior published data with toxigenic C. difficile evaluated in similar conditions. These discussions have been condensed by 35%. In terms of the factors that may explain the dominance of NTCD-E4 we have discussed the following areas: nutrient competition in terms of spore activation and also support of vegetative bacterial populations. Bacteroicins/diffocins are considered and phenotypic analysis as future work suggested to corroborate suggested. Strain specific colonisation factors, immunogenicity, variations in germinant receptors are considered. Antimicrobial susceptibility profile of NTCD-E4 considered as a competitive advantage for NTCD-E4. We appreciate that we can only speculate on some of these areas and that more experimental work is needed in order to determine their relevance in relation to NTCD-E4 and its prevention of RT027 spore activation.

Reviewer 2 Report

General comment.

The manuscript is badly formatted and reader-unfriendly. It took a great effort to go through this. Moreover, it is unduly long. The authors should decrease the length of the manuscript and, also, they must make use of further sub-sections in the text to make reading easier.

As it is now, the manuscript is heading for rejection simply because of its awkward presentation.

Abstract (200 words). Is this really necessary to mention the number of words? (which is wrong by the way….. The correct number is 219 words, if the authors really wish to mention that figure).

Introduction

This is OK in general, although it could have been a bit shorter.

Please describe the objectives of the study clearly in a separate paragraph.

M & M

Please justify why these two particular isolates were used in the study.

Antimicrobial susceptibility testing. Antimicrobial susceptibility testing for the antimicrobials: cefotaxime, ciprofloxacin, ampicillin and 133 tetracycline was performed using a modified agar incorporation method as previously described 134 [45], in order that Brazier’s breakpoint agars could be produced to differentiate between NTCD-E4 135 and RT027, total viable counts and spore counts in mixed culture. A very long sentence that does not help readers. Please break in two or three shorter ones.

Results

Please provide full details of the WGS in supplementary material.

The figures should be reformatted and graphs within the same figure should be placed side-by-side to allow easier comparison.

Discussion

This section is extremely difficult to read. As a first step, it should be divided into sub-sections.

Also, some of the text in Conclusions should be transferred to discussion, as it introduces new ideas.

Conclusions

Please do not introduce new ideas in this section. See above regarding moving text.

Author Response

Thanks for your assessment of the abstract length. We have modified the abstract to simplify and align to the restructured manuscript.

Materials and Methods

Please justify why these two particular isolates were used in the study.

Antimicrobial susceptibility testing. Antimicrobial susceptibility testing for the antimicrobials: cefotaxime, ciprofloxacin, ampicillin and 133 tetracycline was performed using a modified agar incorporation method as previously described 134 [45], in order that Brazier’s breakpoint agars could be produced to differentiate between NTCD-E4 135 and RT027, total viable counts and spore counts in mixed culture. A very long sentence that does not help readers. Please break in two or three shorter ones.

Response – The RT027 isolate selected in this study was one of the original NAP1/BI isolates that was implicated in the US outbreaks of RT027 CDI in 2005; this isolate has been studied in multiple CDI studies previously and produces high-level cytotoxins, bimodal growth, high sporulation and was therefore a significant challenge for the NTCD. NTCD-E4 was selected due to its antimicrobial susceptibility profile allowing for it to be easily viable counted from a mixed culture of C. difficile, i.e. RT027 mixed with NTCD-E4. Additionally, NTCD-E4 is a well-studied isolate in our laboratory in terms of its sporulation, therefore was adjudged a suitable strain to compete against RT027.

Results

Please provide full details of the WGS in supplementary material.

Response – We have added the following statement to the results section: Raw sequence data, as well as the assembled C. difficile NTCD-E4 genome sequence, is available under BioProject number PRJNA917292

The figures should be reformatted and graphs within the same figure should be placed side-by-side to allow easier comparison.

Response - Thanks for this comment – we have reformatted the Figures and added a table to simplify a little for you and we believe this improves the format of the paper.  We have moved the singly-inoculated C. difficile gut model experiments into a supplementary information section so hopefully this reduces the amount of data to consider in the first instance.

Discussion

This section is extremely difficult to read. As a first step, it should be divided into sub-sections.

Response – In line with your suggestion, we have sub-divided the discussion into headed sections, which hopefully makes this simpler to read. We have refined the length of the discussion to remove a little of the depth (>35%) of explanation and comparison to prior studies. We hope that this makes the discussion simpler to read and digest.

Also, some of the text in Conclusions should be transferred to discussion, as it introduces new ideas.

Response - the final section of the conclusions has been moved into the discussion section

Conclusions

Please do not introduce new ideas in this section. See above regarding moving text.

Round 2

Reviewer 2 Report

I still have some concerns about the selection of isolates, so the authors should improve their justification and expand this passage in the revised version.

Author Response

Thank you for your comments, we have strengthened the section on justification of the selection of C. difficile isolates in the methods section and also elaborated in the dicsussion section. As the reviewer does not specifically outline their concerns it is difficult for us to directly address them, but we hope we have achieved this by by the amendment.